# EVTAR: End-to-End Try on with Additional Unpaired Visual Reference

## Abstract

We propose **EVTAR**, an End-to-End Virtual Try-on model with Additional Reference, that directly fits the target garment onto the person image while incorporating reference images to enhance try-on accuracy. Most existing virtual try-on approaches rely on complex inputs such as agnostic person images, human pose, densepose, or body keypoints, making them labor-intensive and impractical for real-world applications. In contrast, EVTAR adopts a two-stage training strategy, enabling simple inference with only the source image and the target garment as inputs. Our model generates try-on results without the need for masks, densepose, or segmentation maps. Moreover, EVTAR leverages additional reference images of different individuals wearing the same clothes to better preserve garment texture and fine-grained details. This mechanism is analogous to how humans consider reference models when choosing outfits, thereby simulating a more realistic and high-quality dressing effect. We enrich the training data with supplementary references and unpaired person images to support these capabilities. We evaluate EVTAR on two widely used benchmarks and diverse tasks, and the results consistently validate the effectiveness of our approach.

## 1 Introduction

Virtual try-on technology aims to generate photo-realistic images of individuals wearing target clothing, with application scenarios including online retail, personalized fashion recommendation systems, and other fields. Despite significant advancements in this domain in recent years, existing methods still suffer from key limitations that hinder their deployment in real-world scenarios. These methods can be broadly categorized into two types: Generative Adversarial Networks (GANs) Goodfellow et al. (2020) and diffusion models Ho et al. (2020); Rombach et al. (2021).

Early studies typically relied on GANs Choi et al. (2021); Han et al. (2018); Wang et al. (2018). They employed warping modules to deform clothing for alignment with the human body, followed by further fusion of the clothing and the human body to achieve visual harmony. However, such GAN-based methods tend to generate unrealistic artifacts, especially when the clothing texture is complex or the human pose is challenging. Recently, research based on latent diffusion models Chen et al. (2024); Xu et al. (2025) has gradually attracted attention in the field of virtual try-on. Their powerful generation and editing capabilities has significantly improved clothing warping performance. Diffusion-based methods Kim et al. (2024a); Zhu et al. (2023); Xu et al. (2024); Choi et al. (2024) significantly address issues related to structural arrangement and texture preservation during the denoising process, which the model requires. Nevertheless, current virtual try-on technologies based on diffusion models generally still rely on additional conditional inputs, such as clothing regions, human poses, key points, and other information.

Despite the remarkable progress of prior virtual try-on approaches, they are still constrained by two critical limitations that hinder their broader applicability: **First**, these approaches rely on multiple external models, such as pose estimatorsGüler et al. (2018) and segmentation modelsKirillov et al. (2023), to process different conditions, which compromises the practicality. Moreover, in practical applications, the quality of conditional inputs—such as the cloth mask—has a substantial impact on the quality of the final try-on results; **Second**, many aspects of clothing, such as style, texture, and detailed design, cannot be fully perceived from the garment image alone; instead, it is more important to consider the overall appearance when the garment is worn by a model. Therefore, in

real-world try-on scenarios, such as online shopping, users are typically more interested in model images rather than the garment itself. They tend to see how the target garment looks when worn on a real person, rather than relying solely on the isolated garment image as a reference. However, existing virtual try-on methods do not support such references due to the lack of corresponding reference data in public datasets.

Based on these observations, this paper proposes **EVTAR**, an end-to-end virtual try-on model that leverages additional reference images. First, EVTAR eliminates the need for auxiliary inputs such as segmentation masks, allowing for simple inference with only a source image and the target garment as inputs. Second, we introduce the use of clothed person images as garment references, which better reflect users' real-world preferences when choosing clothes and enable the preservation of fine garment details that existing methods cannot achieve. Finally, we construct a new dataset with supplementary reference images for training, which empowers EVTAR to achieve both simplified clothing replacement and higher-quality generation results.

In summary, the main contributions of this work are as follows:

- We propose **EVTAR**, a virtual try-on framework that transfers garments onto humans with or without auxiliary conditional information such as segmentation masks or dense pose. This design offers greater flexibility, significantly simplifies the model architecture, and enhances practicality for real-world applications.
- We introduce a novel method that incorporates reference model images for virtual try-on, which greatly enhances the authenticity and the quality of the try-on results. And we construct a high-quality dataset with supplementary reference images and unpaired person images to support our method.
- We provide a virtual try-on model that achieves state-of-the-art (SOTA) performance in both quantitative and qualitative evaluations, and also works well for in-the-wild person-cloth images, demonstrating the excellent capabilities of EVTAR.

## 2 RELATED WORKS

### 2.1 GENERATIVE MODEL VIA FLOW MATCHING

Generative modeling has advanced rapidly, with diffusion models (DMs) Sohl-Dickstein et al. (2015), score-based generative models (SGMs)Song et al. (2021), and flow-based methods emerging as leading paradigms. Flow-based methods have evolved to address inefficiencies in traditional continuous normalizing flows (CNFs)—which require expensive backpropagation through ODE solvers during training Chen et al. (2018)—with a breakthrough being Flow Matching (FM) Lipman et al. (2022), which learns a time-dependent vector field to deterministically transport a simple prior distribution to the target data distribution, using a simulation-free training objective (unlike CNFs) that avoids numerical integration at training time to reduce computational overhead, and achieving comparable or superior sample quality to DMs with far fewer sampling steps (by directly parameterizing the probability flow instead of learning a stochastic reverse process Lipman et al. (2022)).

### 2.2 DIFFUSION-BASED VIRTUAL TRYON

Diffusion models have advanced rapidly in recent years Ho et al. (2020); Rombach et al. (2021), leading to a wide range of diffusion-based approaches in the virtual try-on domain Kim et al. (2024a); Zhu et al. (2023); Choi et al. (2021). Stable Diffusion Morelli et al. (2023); Kim et al. (2024a), with its flexible inpainting and text-guided capabilities, has become widely adopted for virtual try-on tasks. DiffusionCLIP Kim et al. (2022) further incorporates CLIP loss to refine the generated images. DCI-VTON Gou et al. (2023) follows a traditional two-stage pipeline, first warping clothing to align with the body and then fusing the warped garment with the person's image. Subsequently, IDM-VTON Choi et al. (2021) introduces a dedicated GarmentNet module that simultaneously guides both the garment structure and appearance, further improving visual realism. Despite these advances, diffusion-based virtual try-on methods still face notable limitations in terms of practical application and technical completeness, leaving room for further research and optimization. In summary, existing diffusion-based virtual try-on methods either (1) rely on excessive auxiliary annotations or (2) lack support for reference images of clothed individuals. Our EVTAR

model addresses these limitations by combining end-to-end diffusion training with reference-aided generation, while supporting both mask-based inpaint or mask-free editing for virtual try-on.

## 3 METHOD

### 3.1 PRELIMINARY

EVTAR is built upon DiT Peebles & Xie (2023), a scalable Transformer architecture for diffusion-based generation. Images are encoded into a latent space via an autoencoder Kingma & Welling (2013) and then patched into tokens Dosovitskiy et al. (2020). The diffusion process Ho et al. (2020) operates on these tokens, with the Transformer consuming noisy tokens and predicting their denoised counterparts.

We consider the problem of generating images conditioned on a unified context embedding $y$, which may encode textual semantics, style information, or other modality-specific control signals. Let $x$ denote the latent image representation obtained from a VAE encoder. The goal of **flux.1** Labs (2024); Labs et al. (2025) is to approximate the conditional distribution $p(x \mid y)$ by learning a time-dependent velocity field $v(x, y, t)$ that transports a sample from a simple prior $p_0(x) = \mathcal{N}(0, I)$ at $t = 0$ to the data distribution $p_{\text{data}}(x|y)$ at $t = 1$. The dynamics of the conditional density $p(x, t|y)$ are governed by the probability flow ODE

$$\frac{\partial}{\partial t} p(x, t|y) = -\nabla_x \cdot \big( v(x, y, t) \cdot p(x, t|y) \big), \quad x_0 \sim p_0, \ x_1 \sim p_{\text{data}}. \tag{1}$$

To estimate $v(x, y, t)$, we train a neural velocity field $v_{\theta}$ with a diffusion-transformer backbone using the flow matching objective (Lipman et al., 2022):

$$\mathcal{L}_{\theta} = \mathbb{E}_{t, x_i, \epsilon, y_i} \Big[ \big\| v_{\theta}(x, y_i, t) - (x_i - \epsilon) \big\|_2^2 \Big], \quad x = (1 - t) x_i + t \epsilon, \tag{2}$$

where $t \sim \mathcal{U}(0, 1)$, $x_i \sim \mathcal{X}_{train}$, and $\epsilon \sim \mathcal{N}(0, I)$. This training procedure encourages the model to learn a velocity field that consistently transports noisy samples toward the data manifold conditioned on $y$, enabling controllable and context-aware image synthesis at inference time.

In the virtual try-on setting, let $x_i$ denote the image of a person wearing the target cloth, and let $y_i$ represent a collection of conditional inputs, including the agnostic person image $a_i$, the target cloth $c_i$, the dense pose $d_i$, and others. Formally, we write $y_i = [a_i, c_i, d_i, \dots]$. The objective is to progressively transform a Gaussian noise sample $\epsilon$ into the target image $x_i$ guided by conditions $y_i$.

### 3.2 END TO END VIRTUAL TRY-ON WITH EXTRA REFERENCE IMAGES

Unlike previous methods that first mask the target cloth region to generate an agnostic image, our objective is to design a model that can directly fit the target cloth onto the target person, without introducing additional conditions such as densepose Güler et al. (2018) or segmentation masks. To this end, we train a diffusion model on pairs of cloth images $c_i$ and person images wearing different clothes $p_i$, as illustrated in Fig. 1. In practice, to enable the model to support both agnostic and unmasked inputs, we train it on agnostic images and person images with equal probability (50% each), as shown in Fig. 1 (a). During training, we adopt *flux-kontext* Labs et al. (2025) as the backbone, freeze its base parameters, and apply Low-Rank Adaptation (LoRA) Hu et al. (2022) onto the MM-DiT block for parameter-efficient fine-tuning, the training objective is same as the flow-matching loss in Equation 2, where agnostic $a_i$ and unmasked person image $p_i$ are used in turn during the training so that the model will support both mask and mask-free virtual-tryon tasks.

The *flux-kontext* provides a flexible image editing architecture that both the noisy data and conditional inputs are converted into embeddings and concatenated along the sequence dimension. To maintain consistency with this design, we similarly convert multiple conditional inputs, $[a_i, c_i, d_i, \dots]$, into a series of image embeddings and concatenate them along the sequence dimension. Nevertheless, the vanilla *flux-kontext* Labs et al. (2025) supports only a single image condition, which is insufficient for virtual try-on tasks that require at least the target person and garment. To address this, we extend the positional indices in rotary positional embeddings Su et al. (2024) used in the DiT blocks. Each embedding is assigned a three-channel index: the first channel, originally

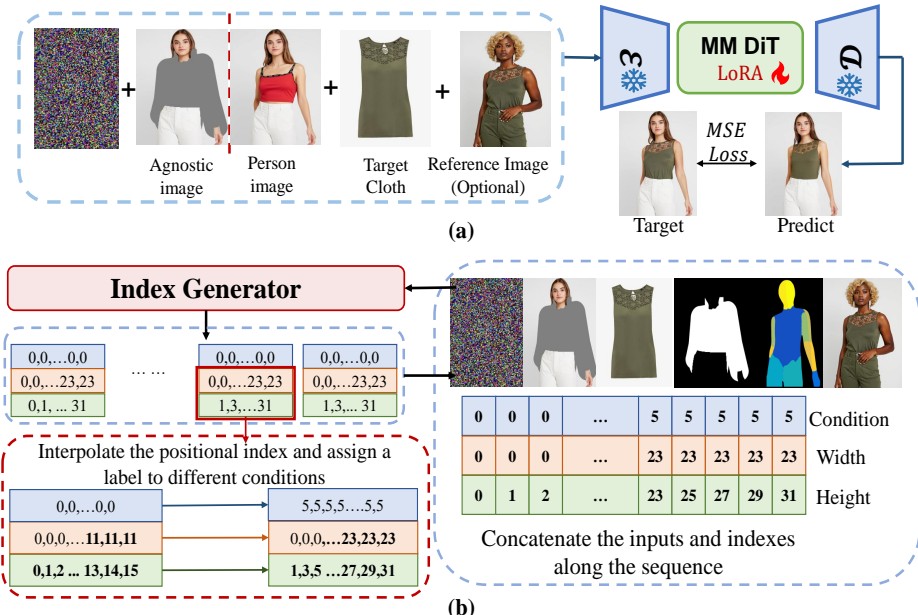

Figure 1: (a) The training pipeline for end-to-end virtual try-on model. (b) Adaptation of the three-channel positional index, where the first channel encodes different conditional inputs, and the second and third channels provide positional information adjusted to the scale of the visual inputs.

0 or 1 to indicate noisy or conditional input, is expanded to values 1-5 to distinguish multiple conditional inputs in our model; the 2nd and third channels encode horizontal and vertical positions into integers, respectively. To support multi-resolution conditions, we scale the positional indices for conditional image embeddings in the second and third channels according to the size ratio between the size of the target image and the conditional images, keeping all indices within a consistent numerical range. We illustrate the modification on the position index of DiT blocks in Fig 1.

Now that we are prepared to train a virtual try-on model. However, we argue that simply relying on cloth images and conditions extracted by external models, as in previous methods, is insufficient to capture the realistic visual effect of fitting a target garment onto a person. A garment's style, texture, and fine design elements are more faithfully represented when worn by a person, whereas viewing the garment in isolation fails to convey these nuances. As shown in Fig. 2, the model cannot accurately identify transparent materials or intricate designs such as lace collars solely from cloth images. In contrast, reference images of human models wearing the garments clearly reveal such details. Therefore, incorporating reference images enhances the model's ability to faithfully transfer fine-grained garment characteristics from the target cloth to the person.

### 3.3 DATA GENERATION FOR TRAINING THE END-TO-END MODEL

To train a model capable of directly taking a person image as input, it is necessary to collect a training set containing unpaired person images of the form "the same person wearing different clothes," denoted as $p_i$, rather than relying on segmentation mask and masked agnostic images. Additionally, to provide reference images for the model, we construct pairs of the form "different persons wearing the same garment," denoted as $r_i$. These two types of data are not available in any existing open-source virtual try-on datasets. In this subsection, we introduce our data engine, which generates the required images, thereby enriching current virtual try-on datasets with supplementary reference and person images and enabling the effective training of our model.

#### 3.3.1 BUILD EXTRA REFERENCE FOR IMAGES

Existing open-source datasets such as **VITON-HD** Choi et al. (2021), **DressCode** Morelli et al. (2022), and **IGPairs** Shen et al. (2025) do not provide the unpaired reference image $r_i$ of "the same

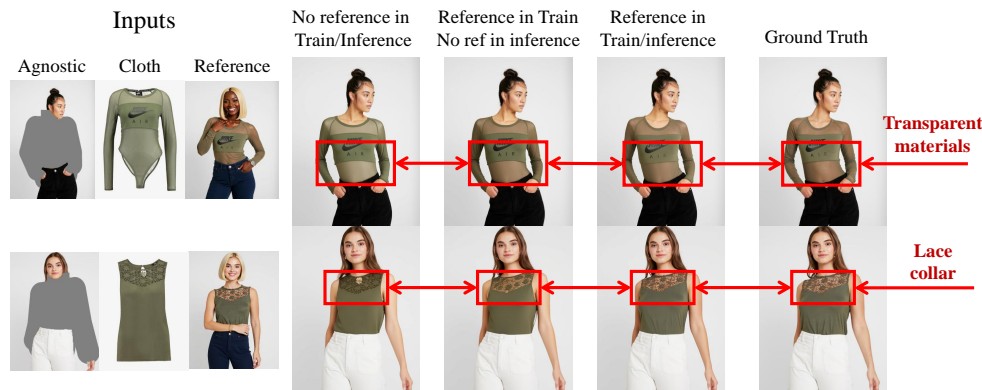

Figure 2: The effect of using reference images for the virtual try-on task. From left to right in the three middle subfigures are: (i) results generated without using reference images during either training or inference; (ii) results generated by a model trained with reference images but inferred without them; (iii) results generated by a model trained and inferred with reference images. Incorporating reference images consistently improves try-on detailed quality and authenticity in training and inference stages. Please zoom in for more details.

person wearing different clothes". We propose generating such reference images using generative models to address this limitation. To construct a high-quality reference set, the generated images should satisfy the following requirements:

1. **Preserve the target garment faithfully.** The color, texture, and design of the target clothing must remain unchanged to ensure accurate reference.

2. **Introduce diversity in person identity.** The person wearing the target garment in the reference image should not be identical to the target person wearing the same garment. Otherwise, the model may overfit to the target image. This diversity can be achieved by altering attributes such as hairstyle, hair/skin color, body pose, or facial expressions.

3. **Vary the non-target garments.** While the target garment remains unchanged, other garments should be modified. For example, if the target garment is an upper-body item, the reference image should retain the same upper-body garment but alter the lower-body clothing, shoes, or accessories.

The overall data generation pipeline is illustrated in Figure 3.

As shown in the pipeline, we employ **Flux-Kontext** Labs et al. (2025) to generate reference images from the target descriptions, leveraging its strong capability to maintain consistency with the input image (1). To address requirement (2), we describe the target image and then intentionally provide an alternative description that ensures the reference image differs from the target input. For this purpose, we use **Qwen2.5-VL** Bai et al. (2025) to generate detailed descriptions of the model's appearance, and instruct it to produce variants that do not resemble the original model at all. Finally, to fulfill requirement (3), we curate a list of garment descriptions representing diverse non-target clothing items. The outlook, action, and outfit descriptions are concatenated as the *positive prompt*, while the original image description is used as the *negative prompt*, and both are fed into the generative model to synthesize reference images.

For each target image, we generate corresponding reference pairs that can be used for both training and evaluation. However, some datasets, such as **FashionTryOn** Zheng et al. (2019) and **IGPairs** Shen et al. (2025), contain numerous duplicated or low-quality samples. To ensure overall data quality, we first apply a filtering process to remove redundant or poor-quality images before generating the final reference data with our data engine.

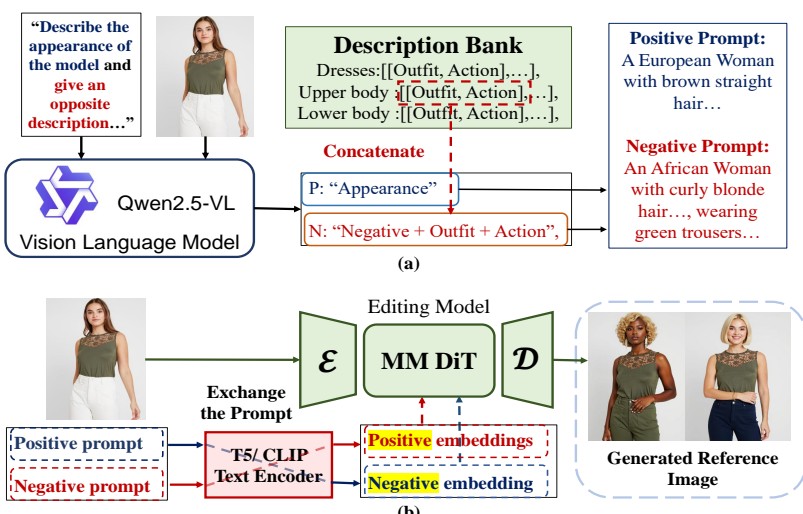

(a)

(b)

Figure 3: The overall pipeline of generating the reference images. We first generate the appearance descriptions using *Qwen2.5-VL* Bai et al. (2025), and then concatenate the appearance with the corresponding actions and outfits to construct the positive and negative prompts, as shown in (a). Subsequently, the images and the textual prompts are fed into the Editing Model, which generates photos of different individuals wearing the same clothes. These results serve as reference images for each image–cloth pair, as shown in (b).

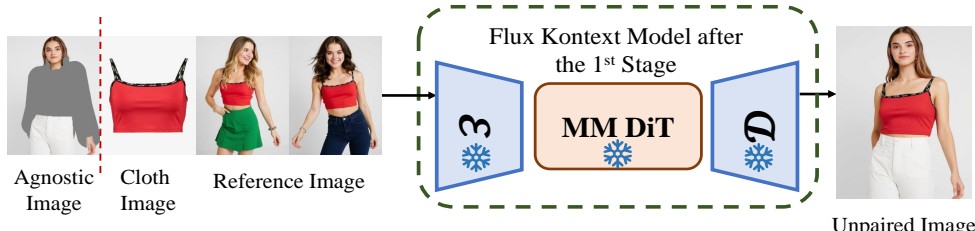

Figure 4: The pipeline of generating the unpaired person image $p_i$. In the first stage, we train a preliminary tryon model with agnostic, cloth, reference image, and extra conditions such as densepose and mask as input. After the training of the first stage, we put the unpaired agnostic-cloth images as input into the model to generate the unpaired person image to train our end to end try-on model.

### 3.3.2 CONSTRUCTION OF THE UNPAIRED PERSON IMAGE DATASET

As shown in Fig. 4, to avoid relying on masks of the target person image and to directly enable our model to take the person image with the target garments as input, we propose to train the model on unpaired person images and force it to predict the same person wearing the target clothes. To generate such unpaired person images, we first train a virtual try-on model using agnostic images. Then, after the first stage, we use this trained model to generate unpaired images from agnostic inputs, as illustrated in Fig. 1 (a), where we use the agnostic image $a_i$, cloth image $c_i$, reference image $r_i$ and extra conditions as input. To ensure consistency, the person–cloth pairs must come from the same subset (e.g., if the agnostic person belongs to the "dresses" subset, the selected garment should also belong to "dresses" rather than "upper body / lower body"). Otherwise, the model would fail to infer the target person wearing the selected garment.

## 4 EXPERIMENTS

We evaluate the effectiveness of our proposed method on two public benchmarks: **DressCode** Morelli et al. (2022) and **VITON-HD** Choi et al. (2021), both composed of images with a resolution of $1024 \times 768$. The VITON-HD dataset consists of 13,670 upper-body image pairs of women, with

Table 1: Quantitative results on the VITON-HD dataset Choi et al. (2021). The best and second best results are reported in **bold** and underline, respectively.

| Method | Input | | | LPIPS$_p\downarrow$ | SSIM$_p\uparrow$ | FID$_p\downarrow$ | KID$_p\downarrow$ | FID$_u\downarrow$ | KID$_u\downarrow$ |
|---|---|---|---|---|---|---|---|---|---|
| | Mask | Pose | Text Prompt | | | | | | |
| CAT-DM Zeng et al. (2024) | ✓ | ✓ | – | 0.0803 | 0.877 | 5.60 | 0.83 | 8.93 | 1.37 |
| IDM-VTON Choi et al. (2024) | ✓ | ✓ | ✓ | 0.102 | 0.870 | 6.29 | – | – | – |
| OOTDiffusion Xu et al. (2025) | ✓ | – | – | 0.0710 | 0.878 | 8.81 | 0.82 | – | – |
| CatVTON Chong et al. (2024) | ✓ | – | – | 0.0565 | 0.870 | 5.425 | 0.411 | 9.015 | 1.091 |
| CatVT2ON Chong et al. (2025) | ✓ | ✓ | – | 0.0570 | 0.8902 | 8.095 | 2.245 | 11.222 | 2.986 |
| OmniVTON Yang et al. (2025) | ✓ | ✓ | ✓ | 0.145 | 0.832 | 7.758 | – | 9.621 | – |
| PromptDresser $_{pose}$ Kim et al. (2024b) | ✓ | ✓ | ✓ | 0.0967 | 0.8778 | 9.07 | 1.16 | – | – |
| PromptDresser Kim et al. (2024b) | ✓ | – | ✓ | 0.1119 | 0.8686 | 8.54 | **0.67** | – | – |
| EVTAR(Ours) | ✓ | – | – | 0.0565 | 0.8734 | 5.4513 | 0.8218 | 8.5772 | 1.0615 |
| EVTAR+ref(Ours) | ✓ | – | – | **0.0489** | **0.8794** | **4.6861** | 0.6760 | 8.4272 | 0.9145 |
| EVTAR mask-free (Ours) | – | – | – | 0.0612 | 0.8658 | 5.9831 | 1.0357 | 8.3974 | 0.8088 |
| EVTAR mask-free + ref. (Ours) | – | – | – | 0.0532 | 0.8720 | 5.1136 | 0.8213 | **8.3222** | **0.7840** |

11,647 pairs for training and 2,032 pairs for testing. The DressCode dataset contains three subsets: upper body, lower body, and dresses, with 48,392 pairs used for training and 5,400 pairs for testing. Since DressCode does not provide cloth-agnostic images, we generate them using the masking tool from . For training, we fine-tune our model using the LoRA technique on the *flux-kontext* backbone, with a rank of 64 and $\alpha$ set to 128. To ensure fair comparison, all images are resized to $512 \times 384$ for both training and inference. In single-dataset experiments, we train the model for 20,000 steps on VITON-HD and 48,000 steps on DressCode, using 8 NVIDIA H200 GPUs.

## 4.1 QUANTATIVE RESULT

We evaluate the numerical results of our virtual try-on model on the VITON and DressCode datasets. As shown in Table 1. Our method (EVTAR) consistently achieves better performance than prior baselines, demonstrating higher try-on fidelity and strong alignment with the target person's pose. With the addition of reference images ("+ref"), performance is further improved, establishing new state-of-the-art results across multiple metrics. Notably, even in the mask-free setting—without agnostic masks or auxiliary inputs—our method maintains garment style correctness and pose consistency, while reaching accuracy on par with or superior to baseline methods, highlighting its robustness and practicality.

Table 2 summarizes the quantitative evaluation on the DressCode dataset. Our method (EVTAR) outperforms all baselines, delivering higher try-on quality and achieving strong consistency with the target person's pose and body structure. The integration of reference images ("+ref") further enhances the results, establishing a new state-of-the-art. Importantly, even in the mask-free setting—without agnostic masks or additional inputs—our method preserves garment styles correctly (e.g., clothing length and design) and maintains high pose alignment, while achieving accuracy comparable to or surpassing prior baselines, demonstrating both robustness and practicality.

## 4.2 QUALITATIVE COMPARISON

As shown in Fig. 5, our method achieves superior visual fidelity compared to all baselines on the VITON dataset. Without relying on reference images, it produces a highly realistic rendering of challenging garment materials such as hollow or semi-transparent fabrics. Moreover, our approach demonstrates the best preservation of garment patterns: the letters on clothes remain clear and consistent with the input, highlighting its advantage in maintaining fine-grained details. With the addition of reference images, the results are further enhanced, and even without using agnostic masks, our method still outperforms most baselines.

As illustrated in Fig. 6, our method achieves more faithful try-on results than all baselines on the DressCode dataset. It handles reflective materials such as leather and metallic fabrics with superior realism, avoiding the over-smoothing or distortion issues observed in other methods. Furthermore, even without providing agnostic masks, our approach can still perform consistent try-on guided by the garment style, correctly preserving length and design without mismatched or inconsistent clothing shapes.

Table 2: Quantitative results on the Dress Code dataset Morelli et al. (2022). The best and second best results are reported in **bold** and underline, respectively. The * marker refers to the results reported in previous work.

| Method | All | | | | | |
|---|---|---|---|---|---|---|
| | $\text{LPIPS}_p\downarrow$ | $\text{SSIM}_p\uparrow$ | $\text{FID}_p\downarrow$ | $\text{KID}_p\downarrow$ | $\text{FID}_u\downarrow$ | $\text{KID}_u\downarrow$ |
| IDM-VTON Choi et al. (2021) | 0.062 | 0.920 | 8.64 | 0.904 | – | – |
| OOTDiffusion Xu et al. (2025) | 0.045 | **0.927** | 4.20 | **0.37** | – | – |
| CatVTON Chong et al. (2024) | 0.0455 | 0.8922 | 3.992 | 0.818 | 6.137 | 1.403 |
| CatVT2ON Chong et al. (2025) | 0.0367 | 0.9222 | 5.722 | 2.338 | 8.627 | 3.838 |
| OmniVTON Yang et al. (2025) | 0.119 | 0.865 | 5.335 | – | 6.450 | – |
| EVTAR(Ours) | 0.0367 | 0.9122 | 3.4783 | 1.1952 | 5.3130 | 1.3552 |
| EVTAR+ref(Ours) | **0.0314** | 0.9180 | **2.9298** | 0.9514 | 5.0703 | **1.1472** |
| EVTAR mask-free (Ours) | 0.0410 | 0.9010 | 3.8390 | 1.3340 | **4.9985** | 1.1675 |
| EVTAR maskfree + ref.(Ours) | 0.0350 | 0.9060 | 3.3360 | 1.1520 | 5.0225 | 1.2819 |

| Method | Upper-body | | | | Lower-body | | | | Dresses | | | |
|---|---|---|---|---|---|---|---|---|---|---|---|---|
| | $\text{FID}_p\downarrow$ | $\text{KID}_p\downarrow$ | $\text{FID}_u\downarrow$ | $\text{KID}_u\downarrow$ | $\text{FID}_p\downarrow$ | $\text{KID}_p\downarrow$ | $\text{FID}_u\downarrow$ | $\text{KID}_u\downarrow$ | $\text{FID}_p\downarrow$ | $\text{KID}_p\downarrow$ | $\text{FID}_u\downarrow$ | $\text{KID}_u\downarrow$ |
| CAT-DM Zeng et al. (2024) | 9.85 | 2.38 | 12.62 | 1.89 | 10.25 | 1.81 | 14.83 | 2.82 | 10.71 | 2.02 | 14.30 | 3.36 |
| OOTDiffusion Xu et al. (2025) | 11.03 | 0.29 | – | – | 9.72 | 0.64 | – | – | 10.65 | 0.54 | – | – |
| PromptDresser Kim et al. (2024b) | 11.00 | 0.74 | – | – | 12.55 | 1.46 | – | – | 11.09 | 1.10 | – | – |
| EVTAR(Ours) | 7.62 | 1.10 | 11.1347 | 0.9779 | 7.60 | 1.38 | 13.0733 | 2.1114 | 7.32 | 1.30 | 11.5621 | 1.9800 |
| EVTAR+ref(Ours) | **6.3867** | **0.8479** | **11.0783** | **0.8710** | **6.6066** | 1.0484 | 12.5593 | **1.6731** | **6.0926** | **1.1636** | 11.1638 | 1.7157 |
| EVTAR mask-free (Ours) | 8.3710 | 1.4330 | 11.2011 | 1.1106 | 8.7850 | 1.5130 | **12.4962** | 1.8307 | 7.36 | 1.33 | 10.7272 | 1.4130 |
| EVTAR maskfree + ref.(Ours) | 7.2030 | 1.1780 | 11.5298 | 1.1234 | 7.8470 | 1.2050 | 12.7370 | 2.0511 | 6.24 | 1.20 | **10.0470** | **1.3047** |

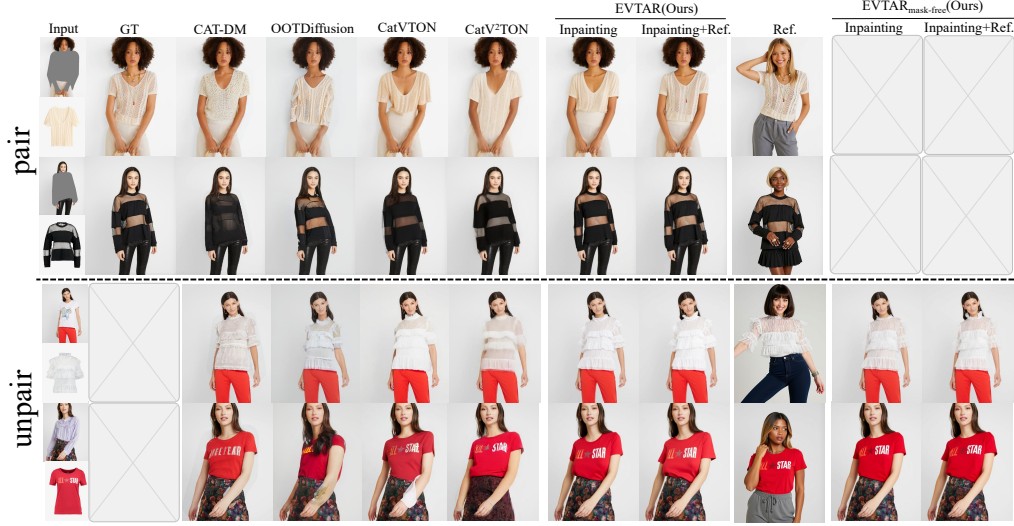

Figure 5: Qualitative comparison on the VITON dataset. "Ref" denotes the reference image used in our method, while "mask-free" indicates that the try-on image is generated conditioned on an unpaired person image instead of an agnostic image.

## 4.3 MIXED DATASET TRAINING AND EVALUATION

To better evaluate the effectiveness of our proposed end-to-end virtual try-on framework and data construction method, we construct a mixed dataset, *Mixed-Virtual-Ref*, by combining data from DressCode, VITON-HD, FashionTryOn Xiao et al. (2025), ViViDFang et al. (2024), and IG-PairsChoi et al. (2024). We carefully select high-quality samples using *Qwen2.5-VL* to filter out low-quality target images, generate cloth-agnostic images with the mask generator from Chong et al. (2024), and produce reference and unpaired person images using our data engine. In total, we collect 103,936 image pairs to train our end-to-end virtual try-on model. The training hyperparam-

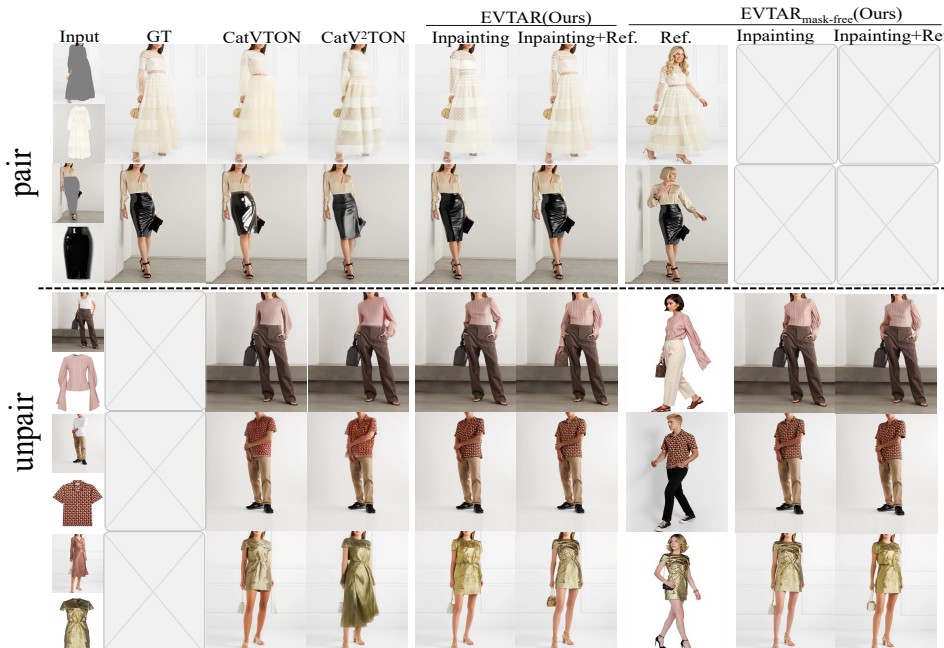

Figure 6: **Qualitative comparison on the DressCode dataset.** "Ref" denotes the reference image used in our method, while "mask-free" indicates that the image is generated using an unpaired person image instead of a masked agnostic image.

Table 3: We compare our approach with Xu et al. (2025) under cross-dataset evaluation, reporting results on both paired and unpaired settings for the VITON-HD and DressCode datasets. The best results are highlighted in **bold**.

| Methods | VITON-HD | | | | | | DressCode | | | | | |
|---|---|---|---|---|---|---|---|---|---|---|---|---|
| | Paired | | | | Unpaired | | Paired | | | | Unpaired | |
| | SSIM↑ | FID↓ | KID↓ | LPIPS↓ | FID↓ | KID↓ | SSIM↑ | FID↓ | KID↓ | LPIPS↓ | FID↓ | KID↓ |
| OOTDiffusion  Xu et al. (2025) | 0.839 | 11.22 | 2.72 | 0.123 | – | – | 0.915 | 11.96 | 1.21 | 0.061 | – | – |
| EVTAR(Ours) | 0.8508 | 6.2313 | 0.7996 | 0.0715 | 9.1128 | 1.0826 | 0.8957 | 3.6982 | 1.1312 | 0.04537 | 5.2208 | 1.2024 |
| EVTAR+ref(Ours) | **0.8593** | **5.1299** | **0.6156** | **0.0597** | 8.5897 | 0.8722 | **0.9026** | **3.1407** | **0.9716** | **0.0377** | 5.0294 | 1.1065 |
| EVTAR mask-free (Ours) | – | – | – | – | 8.8789 | 0.8202 | – | – | – | – | 5.0341 | 1.2341 |
| EVTAR mask-free + ref. (Ours) | – | – | – | – | **8.3905** | **0.6508** | – | – | – | – | **4.8684** | **1.1011** |

eters are kept consistent with those used in the DressCode and VITON-HD experiments. We also evaluate the model on the DressCode and VITON-HD test sets, with quantitative results reported in Table 3. Our method (EVTAR) consistently achieves superior or comparable performance to OOT-Diffusion Xu et al. (2025) across both VITON-HD and DressCode benchmarks under paired and unpaired settings. Notably, the unified model, despite not being trained specifically on DressCode, surpasses the DressCode-specialized baseline in most metrics (e.g., FID and LPIPS), demonstrating strong generalization capability. These results highlight the effectiveness of our approach in learning transferable representations that maintain robustness across diverse datasets.

## 5 CONCLUSION

This paper introduces **EVTAR**, an end-to-end virtual try-on model that supports mask-free inference and leverages additional reference images to enhance the authenticity, realism, and consistency of virtual try-ons. EVTAR achieves state-of-the-art performance in both agnostic and person-specific settings. To enable this, we collect training data from public datasets and generate reference images for each cloth–model pair. Our model supports multi-condition inputs and produces virtual try-on results that faithfully preserve translucent fabrics, intricate designs, and fine-grained details, consistently outperforming existing approaches both quantitatively and qualitatively.

ETHICS STATEMENT

This work adheres to the ICLR Code of Ethics.[1] Our research does not involve human subjects, sensitive personal data, or identifiable private information. The datasets used in our experiments consist of two parts: (1) publicly available datasets that are widely used in the literature, with all usage in compliance with their respective licenses, and (2) internal datasets that are also used in accordance with proper licensing. We do not foresee direct harmful use of our methodology, but we acknowledge that, as with many machine learning techniques, there exists a potential for misuse in unintended application domains. We encourage responsible usage and further discussions within the community regarding broader social and ethical impacts.

REPRODUCIBILITY STATEMENT

We have taken several steps to ensure reproducibility of our results. All model architectures, training settings, and evaluation protocols are described in detail in Section 4. We will release all code and trained models in the future to enable reproduction of our main experiments and to support further research based on our work.

---

[1]https://iclr.cc/public/CodeOfEthics

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

# A    APPENDIX

## LLM USAGE STATEMENT

In accordance with the ICLR policy on the use of large language models (LLMs), we disclose the role of LLMs in the preparation of this work. LLMs were used solely for translation (Chinese to English) and for language polishing to improve the readability and grammar of the manuscript. They were not involved in ideation, dataset creation, model design, or experiment execution. We emphasize that the technical contributions, including the use of FLUX.1dev text encoders in our virtual try-on framework are entirely part of the research methodology and do not constitute LLM-assisted writing. The authors take full responsibility for the correctness and originality of the content presented in this paper.

