# OpenReview forum: "EVTAR: End-to-End Try on with Additional Unpaired Visual Reference"
_ICLR.cc/2026/Conference — ICLR 2026 Conference Withdrawn Submission_

### Official Review · Reviewer_rdLU · 2025-10-18

**Soundness:** 1
**Presentation:** 2
**Contribution:** 1
**Rating:** 2
**Confidence:** 5

**Summary:**

This paper introduces EVTAR, an end-to-end virtual try-on model designed to directly fit a target garment onto a person's image while incorporating additional visual reference images to enhance try-on accuracy and realism.

**Strengths:**

- EVTAR adopts a two-stage training strategy, enabling simple inference with only the source image and the target garment as inputs. It supports both Mask-based and Mask-free virtual try-on, which enhances practicality.

- The model utilizes reference images of different individuals wearing the same clothes  to better preserve garment texture and fine-grained details (such as translucent fabrics and intricate designs). Experiments consistently validate that this mechanism enhances try-on quality.

- The authors construct a new dataset with supplementary reference images and unpaired person images to support the model's end-to-end and reference-aided generation capabilities.

**Weaknesses:**

1. Data Construction and Reference Image Accuracy. The generation of reference images heavily relies on the fidelity capabilities of the base model, Flux-Kontext, particularly in transferring identity and pose while preserving garment details. However, no special enhancement for pose and model transfer is mentioned, raising doubts about the accuracy of the generated reference images. The paper lacks quantitative evaluation or ablation studies on the quality of the generated reference images. If the generated reference figure has color shifts or texture distortion, it could introduce incorrect reference information, potentially leading the try-on model to learn inaccurate fitting effects.

2. Model Innovation and Structural Similarity. The innovation in EVTAR is primarily engineering and application-oriented. The modification to the PE has been explored in other models like OmniControl. Similarly, the concatenation of multiple reference images to guide generation resembles approaches in OmniControl2 and EasyControl. Furthermore, the entire framework is built upon LoRA fine-tuning of FLUX. Overall, the model's methodology shows a high degree of similarity to common diffusion-based editing methods and lacks deep structural or training strategy novelty specific to the virtual try-on domain.

3. Insufficient Ablation and Comparative Experiments. Many existing VTON works (e.g., OmniVTON, CatVTON, cited in the paper) support Person-to-Person (P2P) try-on, which also utilizes a target person's image as a reference. The paper fails to provide ablation studies that compare the effect of using only  the synthesized reference image which is the same setting with  standard P2P VTON which may have the ability to alleviate the detail issues discussed in Figure 2. Furthermore, the number of compared methods is limited, and the authors should include comparisons against other recent and architecturally similar DiT-based methods such as Any2AnyTryon, DreamFit, FitDiT, or IC-LoRA.

4. Contradiction between SOTA Performance and Utility Claim. The quantitative results (Table 1 and 2) clearly indicate that the Mask-based mode combined with the reference image (EVTAR+ref) achieves the highest metrics. The Mask-based model requires an "Agnostic image" as input. Generating this input in an in-the-wild scenario necessitates a segmentation model (i.e., masks) to remove the original garment. Therefore, the achievement of the paper's best performance still relies on external pre-processing tools, directly contradicting the first contribution claim of transferring garments "without auxiliary conditional information such as segmentation masks or dense pose".

**Questions:**

1. Quality Validation and Ablation for the Reference Image: It is suggested to provide a quantitative evaluation of the quality of the generated reference image. For example, calculate similarity between the garment in reference image and the original GT to demonstrate that Flux-Kontext can faithfully preserve garment details and texture while changing the person's identity.

2. Comparison and Necessity of Person-to-Person (P2P) Try-On: Given that many baseline methods (e.g., OmniVTON, CatVTON) support P2P try-on, which is essentially a form of person-to-person reference, please add P2P comparative experiments and ablation studies. Compared to modifications of RoPE in methods like OmniControl, does EVTAR's three-channel PE offer a more fundamental optimization for multi-image fusion? Please provide corresponding technical details or ablation studies to support this claim.

3. Model Innovation and Update of Comparative Methods: Given the similarity in model architecture (DiT/FLUX + LoRA + multi-condition), please add quantitative and qualitative comparisons with the latest VTON/editing methods that are also based on the DiT/FLUX architecture (e.g., Any2AnyTryon, DreamFit, FitDiT, IC-LoRA).

4. Clarification on Optimal Performance Implementation: Explain the reason why the mask-free performance is weaker than the mask-based version, and revise the wording regarding the paper's contributions.

---

### Official Review · Reviewer_HKx3 · 2025-10-30

**Soundness:** 3
**Presentation:** 2
**Contribution:** 2
**Rating:** 2
**Confidence:** 5

**Summary:**

This paper proposed EVTAR, which is an end-to-end virtual try-on framework that transfers garments onto human images using only a source person image and a target clothing image, eliminating the need for auxiliary inputs like segmentation masks or dense pose. By incorporating reference images of different people wearing the same garment, EVTAR preserves fine-grained clothing details and enhances realism. Trained with a novel dataset of unpaired person and reference images, EVTAR achieves state-of-the-art performance on benchmarks like VITON-HD and DressCode, even in mask-free settings.

**Strengths:**

1. EVTAR eliminates the need for auxiliary inputs such as segmentation masks or dense pose, requiring only a source person image and a target garment image, making it more practical for real-world deployment.

2. By leveraging reference images of different individuals wearing the same garment, EVTAR better preserves fine-grained clothing details like texture, transparency, and intricate designs, resulting in more realistic try-on outcomes.

3. The model supports both masked and mask-free inputs, offering greater flexibility and robustness across diverse application scenarios and data conditions.

4. The paper introduces an automated method to generate unpaired person images and reference images using generative and vision-language models, effectively addressing data scarcity in virtual try-on training.

**Weaknesses:**

1. The mask-free paradigm has not been architecturally novel since WUTON and PFAFN, yet valuable work within this framework remains commendable. This paper, however, appears to be a straightforward adaptation of virtual try-on to the powerful FlUX.1-Kontext backbone; the strong quantitative metrics and visually appealing qualitative results largely stem from FlUX.1-Kontext’s inherent generative strength, as evidenced by the authors’ mere LoRA fine-tuning. Consequently, I regard the paper’s contribution as limited.

2. Index Generator is convenient and practical from an engineering perspective, but algorithmically it is not a novel contribution; rather, it is a minor tweak that lets the existing DiT framework ingest multiple conditional images at once.

3. The paper lacks a detailed analysis of training and inference time, memory usage, or comparison with other methods in terms of efficiency, which limits understanding of its scalability.

**Questions:**

1. Your superior quantitative and qualitative results are largely attributable to the advanced Flux Kontext backbone; to demonstrate the intrinsic effectiveness of your proposed method, you should also conduct experiments with a basic backbone such as SD1.5.

2. Why was Qwen2.5-VL chosen, and what considerations guided its selection for generating detailed descriptions?

3. According to Table 1, neither pose nor text prompts are used in your mask-based architecture; how, then, can the model guarantee anatomically correct outputs when confronted with extremely complex human poses?

4. No ablation study is provided to adequately demonstrate the effectiveness of each proposed component.

---

### Official Review · Reviewer_G4xv · 2025-10-30

**Soundness:** 2
**Presentation:** 3
**Contribution:** 2
**Rating:** 2
**Confidence:** 4

**Summary:**

The paper proposes EVTAR, an end to end virtual try on system that works with or without masks and can optionally use reference photos of other people wearing the same garment. The core generator is a diffusion transformer trained with a flow matching objective, adapted with LoRA, and conditioned on multiple visual inputs through a shared indexing scheme. EVTAR reports improvements on VITON-HD and DressCode, and the reference option further boosts detail fidelity in lace, transparency, and text patterns.

**Strengths:**

The strengths are listed as follows.

- (1) **Practical goal.** Inference with only a person image and a product image is aligned with e-commerce use. The optional reference input is a sensible addition that may improve fine texture and transparency.

- (2) **Clean generator design.** A diffusion transformer with flow matching and a single sequence of multi condition inputs keeps the system simple and removes specialized sub modules such as pose estimation and cloth segmentation.

- (3) **Consistent gains.** Within the reported settings, EVTAR and EVTAR with references improve LPIPS, FID, and DISTS against several recent diffusion baselines.

**Weaknesses:**

The major weaknesses are listed below.

- **W1. Application scenario claim is overstated.** The paper argues that existing work requires masks or pose, which is impractical. In fact, there are recent in-the-wild systems that do not require masks and can transfer clothing between different identities without in shop constraints, for example TryOnDiffusion (CVPR 2025). The paper should compare directly to such methods in a mask free setting and reconsider the framing.

- **W2. Odd use of opposite descriptions.** The method asks to describe the appearance of the model and also provide an opposite description. This is counter intuitive for try on, where clothing has concrete attributes. This needs clarification and ablation.

- **W3. Reference pathway under analyzed.** The paper does not quantify how reference identity distance affects results, or how sensitive the method is to reference quality. Results should be bucketed by reference quality and pose gap, and include a study with real catalog references rather than synthetic ones.

- **W4. Narrow related work and limited citations.** The references include only one ICLR and one NeurIPS paper that are closely related, which may not sufficiently situate the work for the ICLR audience.

- **W5. Data and reproducibility gaps.** The synthetic data engine, filtering criteria, and any de-duplication steps are not described with enough detail for reproduction. There is no public release plan, license notes, or leakage checks such as nearest neighbor analysis in CLIP space.

- **W6. Deployment metrics.** The paper does not report inference speed, memory use, or sampling steps, which are critical for an application aimed at e commerce.

**Questions:**

Q1. How exactly are the appearance description and the opposite description created and used. What is the training and inference signal that makes the opposite description helpful for try on. Please add an ablation that removes the opposite description and report the change.

Q2. Which recent mask free did you tune and compare against. If none, can you add direct comparisons to TryOnDiffusion [A] and Person-to-Person Try-On models on the same splits and resolutions.

[A] TryOnDiffusion: A Tale of Two UNets. CVPR, 2023.

[B] ViTon-GUN: Person-to-Person Virtual Try-on via Garment Unwrapping. IEEE Transactions on Visualization and Computer Graphics, 2025.

Q3. What proportion of test references are real human photos versus synthetic images. Please report separate metrics and a small user study for the real reference condition.

Q4. Please report wall clock latency, GPU memory, and step counts for 512×384 and 1024×768, and compare to the fastest strong baseline at matched quality.

---

### Official Review · Reviewer_ZF36 · 2025-10-31

**Soundness:** 2
**Presentation:** 1
**Contribution:** 2
**Rating:** 2
**Confidence:** 4

**Summary:**

This paper presents EVTAR, an end-to-end diffusion-based virtual try-on framework that incorporates additional reference images of people wearing the same garment to improve texture preservation and realism. The approach removes the need for masks or pose inputs and relies on a data generation pipeline to create unpaired and reference samples. Experimental results on VITON-HD and DressCode show competitive performance compared to recent diffusion-based baselines. Overall, the work is conceptually reasonable and technically complete, though the novelty mainly lies in integrating existing components rather than introducing a fundamentally new mechanism.

**Strengths:**

1. The paper tackles a practical limitation of existing virtual try-on methods by enabling mask-free, end-to-end inference, improving usability in real-world scenarios.

2. The idea of using reference images of other people wearing the same garment is intuitive and enhances texture and style consistency in the generated results.

3. The proposed data generation pipeline for synthesizing unpaired and reference images is creative and helps address dataset limitations.

4. Experimental results on VITON-HD and DressCode demonstrate competitive or superior performance compared to several strong diffusion-based baselines.

**Weaknesses:**

## Main Concerns
1. Regarding the visual results, the authors are suggested to mark out the artifacts part of other methods via a bounding box or dashed lines, which would help the readers to have a quick comparison.

2. Also, VTON, as a generation task, more visual comparisons are expected in the appendix or supplementary materials. While this part is missing.

3. There is no discussion about the limitations of the proposed method.

4. As one of the important contributions, the dataset construction, it would be better to include some data samples at least in the supplementary materials, while this is missing.

5. The methodological novelty is limited — the approach mainly combines existing diffusion architectures and LoRA fine-tuning, without introducing new mechanisms or learning objectives.

6. The data generation process for reference and unpaired images lacks clear validation or quality control, making it hard to assess the realism and reliability of the synthetic data.

7. The paper lacks ablation studies to isolate the contribution of reference images or to compare different reference sources (e.g., real vs. generated).

8. The technical description is sometimes vague, especially in Sections 3.2–3.3, where important implementation details and architectural clarifications are missing.

9. There is no analysis of efficiency or inference cost, despite the claim of a simplified, end-to-end design.



## Minor Concerns:
1. Fig.6 needs more careful organization regarding the layout, for example, (i) each subfigure sample is not well aligned; (ii) there is overlap between the top text and the images. This issue can also be found in Fig.5

2. The code is suggested to be released in the future.

3. In Fig.3 and Fig.4, the presentation style of the encoder ($\mathcal{E}$) is not consistent.

4. In Line 126, there should be one ":" after "ODE"

5. In Line 49, there should be a space before the reference.

6. In Fig.3, there are overlapping lines.

**Questions:**

Thank you for the authors' contribution. Below are my main questions:

1. Could the authors provide quantitative or visual ablations to demonstrate the actual contribution of the reference images? For instance, what is the performance gap between using real, generated, or no-reference images?

2. How is the quality of the generated reference and unpaired data validated? Are there any filtering or human verification steps to ensure realism and identity diversity?

3. Since the method claims to be end-to-end and efficient, could the authors report inference time and GPU cost compared to prior approaches such as CatVTON or OOTDiffusion?

4. In the dataset construction process, are there any examples or statistics of the generated data distributions that could be shared (e.g., identity diversity, clothing types)?

5. How does the method handle failure cases or limitations, such as occlusions, complex poses, or unseen clothing types?

Please also refer the "Weaknesses" regarding my main concerns

---

### Note · Authors · 2025-11-14

I have read and agree with the venue's withdrawal policy on behalf of myself and my co-authors.